# Bound-State Electron Dynamics Driven by Near-Resonantly Detuned Intense and Ultrashort Pulsed XUV Fields

**Alexander Magunia \*, Lennart Aufleger, Thomas Ding, Patrick Rupprecht, Marc Rebholz, Christian Ott \***  **and Thomas Pfeifer \***

Max-Planck-Institute for Nuclear Physics, Saupfercheckweg 1, 69117 Heidelberg, Germany;
lennart.aufleger@mpi-hd.mpg.de (L.A.); thomas.ding@mpi-hd.mpg.de (T.D.);
patrick.rupprecht@mpi-hd.mpg.de (P.R.); marc.rebholz@mpi-hd.mpg.de (M.R.)
\* Correspondence: alexander.magunia@mpi-hd.mpg.de (A.M.); christian.ott@mpi-hd.mpg.de (C.O.);
  thomas.pfeifer@mpi-hd.mpg.de (T.P.)

**Abstract:** We report on numerical results revealing line-shape asymmetry changes of electronic transitions in atoms near-resonantly driven by intense extreme-ultraviolet (XUV) electric fields by monitoring their transient absorption spectrum after transmission through a moderately dense atomic medium. Our numerical model utilizes ultrashort broadband XUV laser pulses varied in their intensity ($10^{14}$–$10^{15}$ W/cm$^2$) and detuning nearly out of resonance for a quantitative evaluation of the absorption line-shape asymmetry. It will be shown how transient energy shifts of the bound electronic states can be linked to these asymmetry changes in the case of an ultrashort XUV driving pulse temporally shorter than the lifetime of the resonant excitation, and how the asymmetry can be controlled by the near-resonant detuning of the XUV pulse. In the case of a two-level system, the numerical model is compared to an analytical calculation, which helps to uncover the underlying mechanism for the detuning- and intensity-induced line-shape modification and links it to the generalized Rabi frequency. To further apply the numerical model to recent experimental results of the near-resonant dressing of the 2s2p doubly excited state in helium by an ultrashort XUV free-electron laser pulse we extend the two-level model with an ionization continuum, thereby enabling the description of transmission-type (Fraunhofer-like) transient absorption of a strongly laser-coupled autoionizing state.

**Keywords:** atomic physics; ultrashort physics; bound-bound electronic transitions; strong-field couplings; transient-absorption spectroscopy; line shape manipulation; free-electron-laser; numerical calculations

## 1. Introduction

The manipulation of electronic states in atoms with intense electromagnetic fields has been studied theoretically for several decades [1,2]. One of the many investigated cases has been the existence of an excited state energetically embedded in an ionization continuum, nowadays known as an autoionizing Fano state [3], for which resonant couplings with strong fields have been studied in detail [3,4]. Also, transient-absorption spectroscopy experiments have been carried out more recently with sensitivity to strong-coupling dynamics of Fano resonances in rare gas atoms [5–9]. These experiments employ broadband extreme-ultraviolet (XUV) attosecond pulses in combination with time-delayed intense femtosecond laser pulses in the near-infra red (NIR) spectral regime. Characteristic absorption lines are observed in the XUV-pulse spectrum after transmission through a moderately dense cloud of atoms or molecules. Hereby, measuring the transient absorption signal allows one to access the real-time dynamics of the XUV-excited system when it is driven by the NIR laser pulse [10] while

retaining state-specific spectroscopic resolution. With the advent of XUV and x-ray free-electron-lasers (FELs) [11,12], new possibilities opened up for ultrafast nonlinear light-matter interaction at high photon energy and intensity [13–15]. Regarding transient absorption spectroscopy with intense XUV-FEL pulses, signatures of strongly driven XUV resonant transitions have been observed [16,17]. Hereby, the intense XUV-FEL laser pulses may strongly couple the ground state directly to a highly excited state of an atom. Predicting the ionization yield and photoelectron emission fur such a transition in helium, XUV-FEL absorption spectroscopy at high intensity has been studied theoretically [18–20]. In this work, we computationally model intense-field XUV absorption spectroscopy by directly observing the intense XUV light after its transmission through a resonant medium and use it to explain and predict line-shape asymmetry changes of resonant transitions strongly coupled by XUV light. To properly account for the broad spectral bandwidth of the ultrashort pulses, we solve the underlying few-level system of resonant couplings numerically and further compare it to an analytical approximation [1,2] for monochromatic fields as a function of the detuning in terms of the generalized Rabi frequency. Our numerical findings for the autoionizing 2s2p doubly excited state in helium further support recent experimental results that have been obtained with intense XUV-FEL pulses [16].

## 2. Methods

First, we consider a two-level system for our numerical model, for which the Hamiltonian $H_{2lvl}$ is described as a matrix with two states at energy $E_g = 0$ eV and $E_e = 60.15$ eV, for the ground and the excited state, respectively, which are coupled in dipole approximation by the dipole matrix element $d_{ge} = d_{eg} = -0.035$ a.u. (a.u. = atomic units; which are used throughout this work unless stated otherwise). The parameters are chosen to match the relative energy separation and dipole coupling strength between the $1s^2$ ground state and the 2s2p doubly excited state in helium [21]. Hereby at first, we explicitly neglect the configuration interaction with the ionization continuum of the 2s2p doubly excited state, but still take its finite autoionization lifetime into account by means of the excited-state decay parameter $h\Gamma_e = 37$ meV. Going one step further, we consider a second model, where we account for the configuration interaction between the excited state and the one-electron ionization continuum by extending our first model to a three-level system with Hamiltonian $H_{3lvl}$, which additionally includes an energetically broad quasi-continuum state that serves as a short-lived ionization-loss channel for the system. The parameters of this quasi-continuum state are set to energy $E_c = 32.654$ eV, decay width $h\Gamma_c = 39.728$ eV, dipole matrix element $d_{gc} = d_{cg} = 0.675298$ a.u. for its dipole coupling to the ground state and configuration interaction $V_{CI} = 0.0373209$ a.u. for the inter-channel coupling between the quasi-continuum state and the 2s2p state. In this three-level system, the excited-state parameters are slightly altered to the excited-state energy $E_e = 60.122$ eV and to the dipole matrix element $d_{ge} = -0.0493158$ a.u. for dipole coupling to the ground state, while the previously introduced direct decay of the excited state can be omitted ($h\Gamma_e = 0$ eV), since its autoionization is now explicitly taken into account through $V_{CI}$. The numerical parameters of this three-level system have been chosen such that the weak-field transient absorption profile of the $1s^2$–2s2p transition agrees with the reported literature values [21], most importantly now also including its asymmetric line shape. See also Figure 1b for a simplified energy-level scheme of the system. The XUV laser pulse is treated as a classical electric field $E(t)$ with a Gaussian envelope of duration $T_{FWHM} = 3$ fs (intensity full width at half maximum). To summarize, the Hamiltonian matrices $H_{3lvl}$ and $H_{2lvl}$ for the three- and two-level system, respectively, are given by:

$$H_{3lvl} = \begin{pmatrix} E_g & d_{ge}\,E(t) & d_{gc}\,E(t) \\ d_{ge}\,E(t) & E_e & V_{CI} \\ d_{gc}\,E(t) & V_{CI} & E_c + i\,\Gamma_c/2 \end{pmatrix}, \text{ and } H_{2lvl} = \begin{pmatrix} E_g & d_{ge}\,E(t) \\ d_{ge}\,E(t) & E_e + i\,\Gamma_e/2 \end{pmatrix}, \quad (1)$$

with the numerical parameters as defined above. The time-dependent Schrödinger equation $i\hbar\frac{\partial}{\partial t}|\Psi(t)\rangle = H(t)|\Psi(t)\rangle$ is solved on a discrete time grid (time steps $\Delta t = 0.1$ a.u. $= 2.42$ as) through numerical diagonalization of the Hamiltonian at each time step. The state vector of

the system, $\left|\Psi(t)\right\rangle^{(2\text{lvl})} = \left[c_g(t), c_e(t)\right]^T$ or $\left|\Psi(t)\right\rangle^{(3\text{lvl})} = \left[c_g(t), c_e(t), c_c(t)\right]^T$, is expressed through the complex-valued coefficients $c_i$ ($i$ = g, e, c). For the initial state, only the ground state is populated ($c_g(t=0) = 1$). The coefficients are propagated in time within the diagonalized Hilbert space by multiplication with a pure phase factor, $e^{-iE_i\Delta t/\hbar}$, containing the eigenvalues $E_i$ $\left[i = 1, 2, \left(3\text{only when } c_c \text{ is included}\right)\right]$ of the diagonalized Hamiltonian. The complex time-dependent dipole moment $d^+(t)$ of the system rotating with positive frequencies (containing terms $\propto e^{i\omega t}$) is given by:

$$d^+(t) = d_{ge}\, c_g(t)c_e^*(t) + d_{gc}\, c_g(t)\, c_c^*(t), \tag{2}$$

with the previously defined dipole matrix elements $d_{ge}$ and $d_{gc}$. The second term in Equation (2) does not contribute for the two-level system. The optical density (OD) of transient absorption is calculated with the frequency spectrum of the time-dependent dipole moment $d(\omega) = \text{FT}\{d^+(t)\}$ obtained after Fourier transformation and is given by taking the negative decadic logarithm of the ratio between transmitted (numerator) and incoming (denominator) spectral intensity:

$$\text{OD}(\omega) = -\log_{10}\left(\frac{\left|E(\omega) + i\,\eta\cdot d(\omega)\right|^2}{\left|E(\omega)\right|^2}\right). \tag{3}$$

here, $E(\omega)$ is the frequency spectrum of the incoming laser pulse and $\eta = 10^{-4}$ is a constant connected to the macroscopic particle density of the target medium. Its numerical value is arbitrary and here chosen small enough to avoid macroscopic propagation effects. Equation (3) contains the term $\left|E(\omega) + i\,\eta\cdot d(\omega)\right|^2$, which is an interferometric superposition of the polarization response ($P = \eta\cdot d$ [22]) and the driving field following Maxwell's equations, which is at the heart of transient absorption spectroscopy (see Figure 1a). The resulting optical density (Figure 1d,e) is, thus, directly sensitive to the relative phase between the driving field and the dipole response [8] manifesting in an absorption profile, which can be parametrized by a generalized Fano line shape. To quantify the asymmetry of the resonant absorption lines we hence fit the OD with a Fano profile:

$$\text{OD}(\omega) = A\frac{(\varepsilon(\omega) + q)^2}{(1 + \varepsilon^2(\omega))} + B, \; \varepsilon(\omega) = \frac{\hbar\omega - E_r}{\Gamma_r/2}, \tag{4}$$

where q is the Fano asymmetry parameter, $\varepsilon(\omega)$ is the relative photon energy depending on the resonance position $E_r$, its spectral width $\Gamma_r$ and the photon energy $\hbar\omega$. A and B are fitting parameters for the amplitude and offset, respectively. In the case of the weak-field two-level system, the absorption profile is unperturbed and assumes a Lorentzian shape (Figure 1d), which is the limiting case of a Fano profile for $q \rightarrow \pm\infty$. When the third quasi-continuum state for the 2s2p resonance in helium is included in the weak-field case, the q parameter equals −2.75 [23]. With strong laser fields and in the "impulsive limit" [8,24,25], i.e., when the pulse duration (here: 3 fs) is short compared to the resonance lifetime (here: $\tau = 1/\Gamma_r = 17$ fs), the absorption profile asymmetry, which is captured by the q parameter, can be modified. For the transmission-geometry transient absorption, this change is related to a phase shift $\Delta\varphi_D$ of the time-dependent dipole moment with respect to the weak-field response of the system [8] and is given by:

$$q\,(\Delta\varphi_D) = -\cot\left(\frac{\Delta\varphi_D}{2}\right) \Leftrightarrow \Delta\varphi_D = 2\arg(q-i). \tag{5}$$

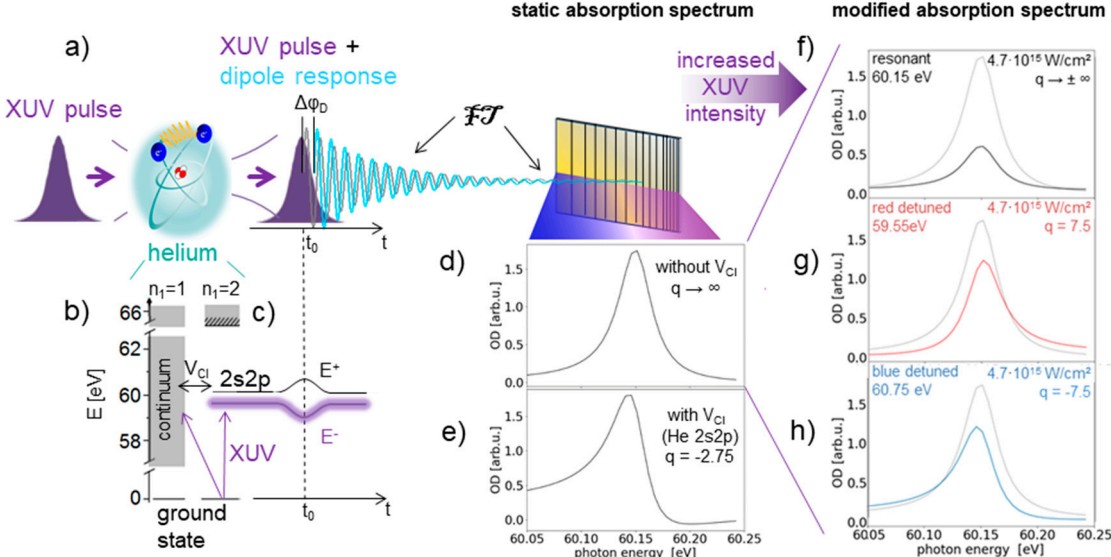

**Figure 1.** (**a**) the interaction of an ultrashort extreme-ultraviolet (XUV) pulse (violet) with an atom, e.g., helium, leads to the emission of a dipole response (blue) interfering with the driving XUV pulse. (**b**) simplified energy level scheme of helium. With a photon energy of 60.15 eV the helium atom can be either singly ionized or excited into a doubly excited state, which autoionizes due to its configuration interaction $V_{CI}$ with the ionization continuum. Corresponding absorption spectra are shown in (**d**,**e**). If the configuration interaction is disregarded, the absorption profile is Lorentzian (**d**), whereas the inclusion of the configuration interaction leads to a Fano absorption line shape (**e**). (**c**) dressed states are generated in the presence of an intense XUV pulse. The states' energy separation increases with increasing XUV laser intensity and depends on the detuning [1,2], therefore their energy shift follows the Gaussian pulse envelope in time (compare to a), here illustrated for a red-detuned case. This leads to a temporal dipole phase shift $\Delta\varphi_D$ with respect to the weak-field response (grey in a) [8]. (**f**)–(**h**) manipulated absorption profiles of the two-level system (without configuration interaction) initiated by the strong coupling with an XUV pulse of different detunings. (**f**) the peak intensity of a resonant laser pulse is increased to $4.7 \times 10^{15}$ W/cm$^2$ leading to a decreased amplitude of the resonance, but leaving the line shape unchanged ($q \to \pm\infty$). (**g**,**h**) the additional detuning of the laser pulse to central photon energies $\hbar\omega_c$ of 59.55 eV or 60.75 eV, leads to modified now asymmetric absorption line shapes with a positive or negative q -parameter, respectively.

## 3. Results and Discussion

### 3.1. Two-Level System

In Figure 1f–h, modified absorption profiles of a two-level system driven by Gaussian laser pulses of 3 fs duration with high peak intensity $I_0 = 4.7 \times 10^{15}$ W/cm$^2$ and with detunings $\Delta/E_e = 0$ (Figure 1f, resonant), $\Delta/E_e = -0.01$ (Figure 1g, red detuned) and $\Delta/E_e = +0.01$ (Figure 1h, blue detuned) are shown in comparison to the unperturbed absorption profile (grey). Hereby the detuning is defined as

$$\Delta = \hbar\omega_c - E_r, \tag{6}$$

where $\hbar\omega_c$ is the central photon energy of the XUV pulse. It should be noted that the corresponding spectral bandwidth of the XUV laser pulse amounts to ~0.6 eV (intensity full width at half maximum), which corresponds to $\lesssim$1% of the central photon energy. The detuning is thus changed only within the spectral bandwidth of the XUV pulse which leads to near-resonant couplings at all detunings. To further investigate the mechanism behind the modified line shapes, we now systematically vary the XUV intensity and detuning. In Figure 2c we plot the fit results of the q parameter for the two-level resonance under the influence of an XUV pulse with peak intensities ranging from $I_0 \approx 0.1 \times 10^{15}$ W/cm$^2$ to $I_0 \approx 4.7 \times 10^{15}$ W/cm$^2$ and detunings ranging from $\Delta = -0.65$ eV to

$\Delta = +0.65$ eV. For low intensities, the resonance is unperturbed and forms a Lorentzian shape, where the fit yields q $\rightarrow \pm\infty$. The divergence of q for a Lorentzian line shape is better quantified with the corresponding dipole phase shift $\Delta\varphi_D$ (see Equation (5)), which is shown in Figure 2b. Instead of the divergent q parameter, the phase shift is now smoothly changing around zero for low intensities. The three horizontal lineouts for detunings of −0.6 eV (red), 0 eV (black) and +0.6 eV (blue) are shown in Figure 2a and the three vertical lineouts for peak intensities of $0.1 \times 10^{15}$ W/cm² (violet), $2.4 \times 10^{15}$ W/cm² (orange) and $4.7 \times 10^{15}$ W/cm² (black) are depicted in Figure 2d.

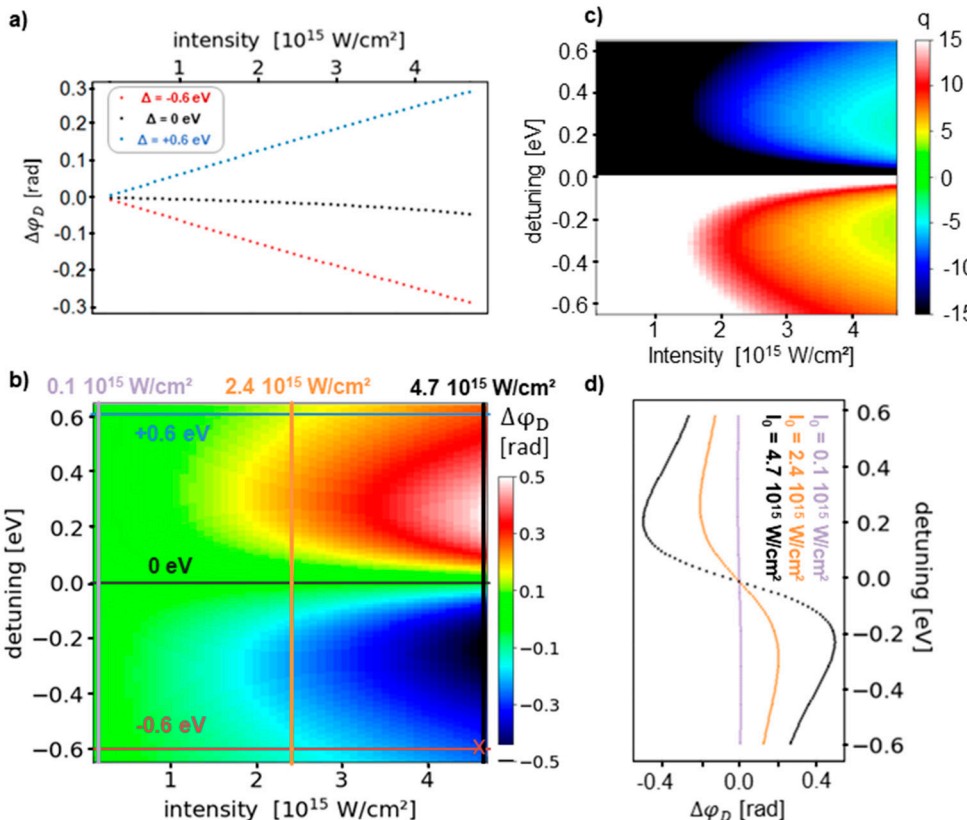

**Figure 2.** Results for the two-level system obtained by fitting a generalized Fano line shape (Equation (4)) to the simulated optical density. (**b**) dipole phase shift $\Delta\varphi_D$ obtained after applying Equation (5) to the direct fit results of q shown in (**c**) as a function of detuning and intensity, with corresponding (**a**) horizontal lineouts as a function of intensity for different detunings and (**d**) vertical lineouts as a function of detuning for different intensities. (**c**) the fitted q -parameter as a function of detuning and intensity.

We conclude from these results, that the dipole phase shift changes monotonically with the laser intensity (decreasing for red detuning and increasing for blue detuning), is equal in amplitude but flipped in sign if the detuning is flipped in sign and approximately vanishes for resonant driving pulses.

### 3.2. Analytical Approximation

In the following, we will relate these findings to the well-known description of a strongly coupled two-level system in the generalized Rabi formalism [1,2]. To understand the connection between the splitting and shifting of the involved energy levels to the modified absorption profile, we propose the comparison to the following analytical approximation: In the presence of the intense XUV pulse, the excited state of the system is split into two states $E^\pm$, which are the eigenstates of the laser-dressed two-level system, and which are energetically shifted with increasing field strength. For a temporally Gaussian-shaped XUV pulse envelope, the dressed states also shift energetically along a Gaussian curve in time (see Figure 1c). Hereby, one dressed state emerges directly from the atomic excited

state, whereas the other one is created from the ground state with the absorption of an XUV photon. It depends on the detuning $\Delta$ of the XUV pulse, which of the two bare states transforms into the energetically upshifted state $E^+$, and which transforms into the downshifted state $E^-$ of the dressed two-level system. For a red-detuned XUV pulse, i.e., when the central photon energy of the pulse is smaller than the resonance transition energy, the excited state translates into $E^+$, whereas for a blue-detuned XUV pulse it translates into $E^-$. After the interaction with the XUV pulse, both bare states have collected a phase shift:

$$\Delta\varphi_{g,e} = \int \Delta E_{g,e}(t)\, dt, \tag{7}$$

due to the energy difference between the laser-dressed and bare states with respect to the weak-field coupling. The energy shifts are thus given by:

$$\Delta E_g(t) = \begin{cases} E^-(t) - E_{g+\omega}\,, \Delta < 0 \\ E^+(t) - E_{g+\omega}\,, \Delta > 0 \end{cases}, \quad \Delta E_e(t) = \begin{cases} E^+(t) - E_e\,, \Delta < 0 \\ E^-(t) - E_e\,, \Delta > 0 \end{cases}, \tag{8}$$

where $E_{g+\omega} = E_g + \hbar\omega_c$ stands for the ground state energy plus the XUV photon energy. With the dipole phase $\varphi_D = \arg(c_g c_e^*)$ according to Equation (2), this directly translates into a phase shift of the resulting dipole emission:

$$\Delta\varphi_D = \Delta\varphi_g - \Delta\varphi_e. \tag{9}$$

For a monochromatic driving field of constant electric-field amplitude $E_o$, we can use the generalized Rabi formulas and obtain

$$E^\pm = E_e + \frac{\Delta}{2} \pm \frac{1}{2}\sqrt{\Delta^2 + |\hbar\Omega|^2} \tag{10}$$

for the energies of the laser-dressed states with the Rabi frequency $\Omega = d_{ge}E_o/\hbar$. We assume a rectangular pulse profile of duration $T = 2.82$ fs for the XUV pulse so that the dipole phase shift can now be analytically written as:

$$\Delta\varphi_D = \frac{(\Delta E_g - \Delta E_e)}{\hbar}T = \frac{1}{\hbar}\begin{cases} \left(-\Delta - \sqrt{\Delta^2 + |\Omega|^2}\right)T\,, \Delta < 0 \\ \left(-\Delta + \sqrt{\Delta^2 + |\Omega|^2}\right)T\,, \Delta > 0 \end{cases}. \tag{11}$$

We note, that this approximation is only valid in the impulsive limit ($T \ll 1/\Gamma_r$). The choice of $T = 2.82$ fs was made to match the integrated intensity –the fluence– of the rectangular pulse with the one of the $T = 3$ fs Gaussian pulse used previously. This analytical calculation of the dipole phase shift already explains some of the previously obtained trends (see Figure 3a): The dipole phase shift increases monotonically with peak intensity and assumes opposite sign for positive/negative detuning. However, we now observe a sharp jump from a maximal to a minimal phase shift across $\Delta = 0$ instead of a smooth transition through $\Delta\varphi_D = 0$ (compare with Figure 2b). This discontinuity in fact arises from the initial assumption of monochromatic light fields and thus well-defined detunings. Note that for zero detuning Equation (11) does not apply and thus the dipole phase shift is set to zero in this illustration. For temporally short and, thus, spectrally broad pulses, all spectral components of the pulse, i.e., photons with a detuning within the spectral pulse band width, contribute to this energy- and phase-shifting mechanism at the same time. Therefore, the resulting dipole phase shift is expected not to be determined by the central photon energy of the driving pulse alone, but rather by the average over its spectrum. In Figure 3b, we convolute (result shown in violet) the analytical expectation (shown in blue) of the detuning-dependent dipole phase shift for lowest intensities, $I_0 = 0.1 \times 10^{15}$ W/cm$^2$, with the spectrum of the 3 fs Gaussian driving pulse, which is also a Gaussian profile with an intensity FWHM of 0.61 eV (shown in red) and compare it to the numerical results from Figure 2d (shown

in black). Additionally, we scale the magnitude of the convoluted dipole phase shift with a factor of ~0.53 for a better qualitative comparison to the numerical results. One possible interpretation for this rescaling with a factor of approximately one half could be that the "effective" phase shift of the dipole emission is mainly induced by the trailing edge of the pulse, while its leading edge is mainly responsible for the excitation of the system. Since the dipole phase shift (see Equation (11)) is of equal magnitude but changes sign for positive/negative detuning, it evaluates to zero after convolution for a resonant driving pulse. In Figure 3c,d, we plot the detuning-dependent dipole phase shift from Figure 3a in the same manner as in Figure 3b, but for higher peak intensities, $I_0 = 2.4 \times 10^{15} \, \text{W/cm}^2$ and $I_0 = 4.7 \times 10^{15} \, \text{W/cm}^2$, respectively. For the result of this convolution, we find excellent agreement of the analytically calculated and convoluted phase shifts with the numerically obtained phase shifts, when we set the FWHM of the Gaussian convolution function to 0.45 eV and 0.3 eV and rescale the convoluted dipole phase shifts by ~0.64 and ~0.65, respectively. The finding of an intensity-dependent FWHM of the convolution function indicates a nonlinear process, which needs further investigation. Yet, this agreement shows how detuning-dependent line-shape asymmetry changes from impulsive dressing at high intensity can be captured and approximated within an analytical framework of the well-known Autler–Townes energy-level splitting of a strongly coupled two-level system using the generalized Rabi formalism.

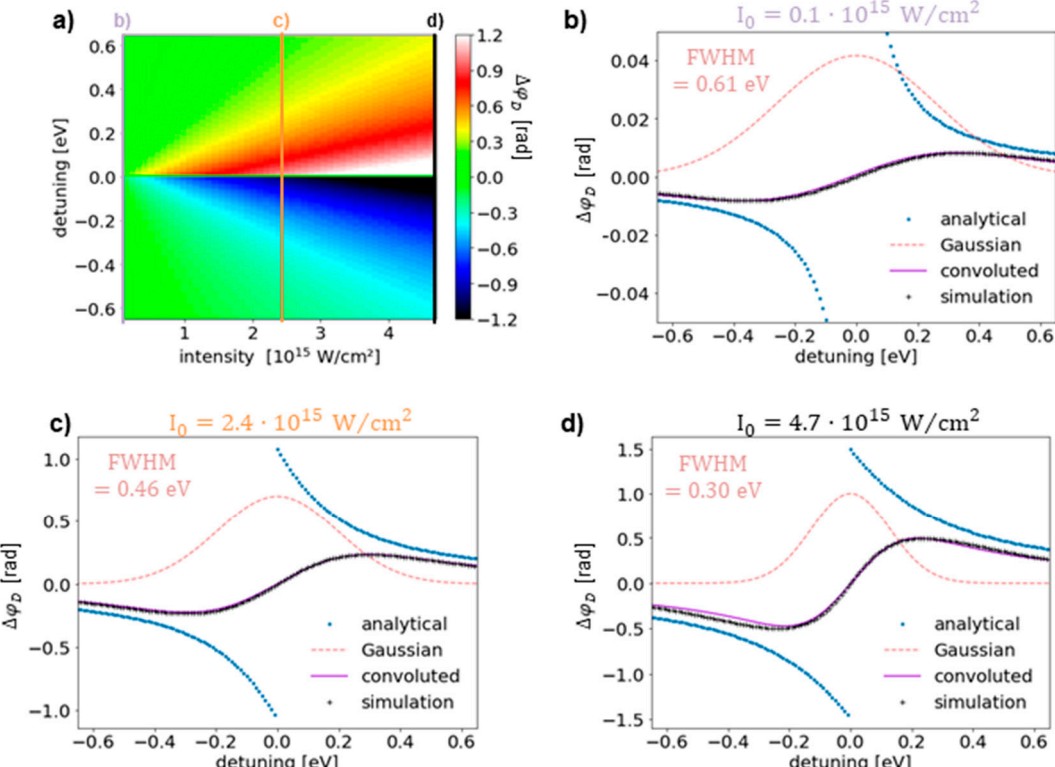

**Figure 3.** (**a**) dipole phase shifts $\Delta\varphi_D$ calculated analytically with Equation (11). A jump from positive to negative dipole phase shifts at zero detuning is clearly present as well as the monotonic increase (decrease) of $\Delta\varphi_D$ for blue (red) detuned pulses. (**b**–**d**) lineouts of the dipole phase shift (of Figure 3a, blue) are convoluted along the detuning axis (violet) with an intensity-dependent Gaussian profile (depicted at the resonance position, red) and compared to the numerical results (Figure 2d), black). (**b**) for the lowest intensity, $I_0 = 0.1 \times 10^{15} \, \text{W/cm}^2$, the Gaussian convolution function with a FWHM = 0.61 eV corresponds to the spectrum of the 3 fs Gaussian driving pulse. (**c,d**) for higher intensities, $I_0 = 2.4 \times 10^{15} \, \text{W/cm}^2$ and $I_0 = 4.7 \times 10^{15} \, \text{W/cm}^2$, respectively, a decreasing FWHM of the Gaussian convolution function provides the best fit of the convoluted analytically calculated dipole phase shifts to the corresponding numerical results.

### 3.3. Autoionizing 2s2p Resonance in Helium

Pertaining to previous experimental results obtained at the Free-Electron Laser in Hamburg (FLASH) [16], we focus in the following on the impact of the detuning for the impulsive dressing of the 2s2p doubly excited state in helium, where the excited state couples to an ionization continuum. By repeating the numerical simulations for the autoionizing state (see $H_{3lvl}$ in Equation (1)) for the same peak intensity and detuning values as used for the two-level system, the dipole phase shift and q parameter are again obtained from the same Fano fitting procedure as in the two-level case and are depicted in Figure 4. The literature asymmetry parameter q = −2.75 is reproduced for the lowest pulse intensity (see Figure 4b) and corresponds to an intrinsic dipole phase shift of $\varphi_0 \approx 0.7$ rad. To quantify only the field-induced dipole phase shift $\Delta\varphi'_D$, we subtract this offset phase: $\Delta\varphi'_D = \Delta\varphi_D - \varphi_0$ for Figure 4a. With increasing pulse intensity, we observe the three following features in the dipole phase shifts: (i) there is no dipole phase shift and hence no modification of the absorption line shape approximately for $\Delta \approx -0.2$ eV, which weakly depends on the laser intensity (violet line in Figure 4a, (ii) the dipole phase shift is negative for $\Delta \lesssim -0.2$ eV and positive for $\Delta \gtrsim -0.2$ eV and (iii) the magnitude of the dipole phase shift increases with the pulse peak intensity. Features (ii) and (iii) are very similar to the previous results of the two-level system, indicating that similar energy and phase shifts take place when the coupling to an ionization channel is explicitly included in contrast to the previous simple decay of an isolated state. However, now we observe a significant offset detuning $\Delta \approx -0.2$ eV for which the measurable line-shape asymmetry is approximately unchanged for increasing laser intensity. Furthermore, the magnitude of the positive and negative dipole phase shift around this offset are also different in strength. We give a first qualitative explanation of these new observations through the initial (i.e., weak-field) asymmetric Fano line shape by invoking a similar idea as for the analytical and convoluted calculations of the two-level system: For the given q-parameter of −2.75, the weak-field absorption cross section of the 2s2p resonance has a peak at photon energies smaller than the resonance position and a minimum for higher photon energies (see Figure 1e). Hence, the spectral components of a resonant driving XUV pulse with lower photon energies are more likely to be absorbed and, thus, their contributions to the dipole phase shift are expected to be more pronounced than the contribution of the higher photon energy components of the pulse. The dipole phase shift, thereby, is not cancelled out for a resonantly tuned XUV pulse as is the case for a weak-field Lorentzian resonance, but rather it is cancelled out for a slightly red detuned pulse, which is in qualitative agreement with the finding of an unchanged line-shape asymmetry around $\Delta \approx -0.2$ eV in Figure 4a. The observed trend, thus, seems to be a direct consequence of the configuration interaction and channel interference of a Fano resonance. We would like to note that the detuning offset ($\Delta \approx -0.2$ eV) for this specific case may depend on the exact description of the internal configuration interaction between the excited state and the ionization continuum. Since we utilize a quasi-continuum state here, we do not expect this value to be quantitatively precise. Nevertheless, the exact model choice of the coupled configurations is not expected to change our main finding of an asymmetric line-shape modification as a function of detuning for the impulsive dressing of an autoionizing state.

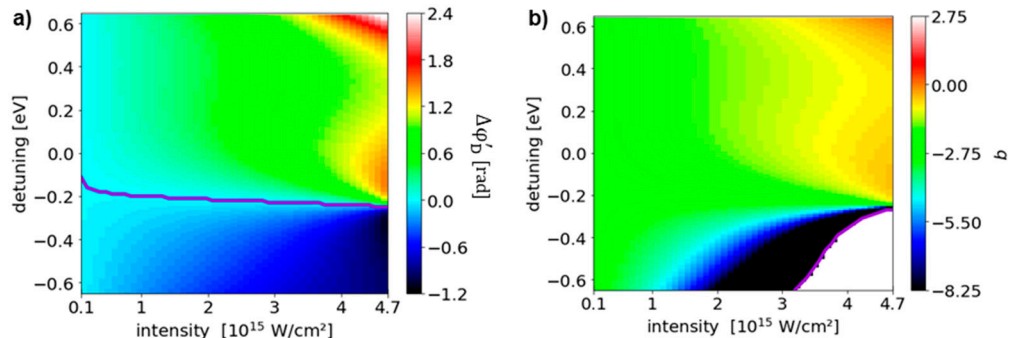

**Figure 4.** (**a**) dipole phase shift $\Delta\varphi'_D$ and (**b**) q parameter of the autoionizing 2s2p resonance in helium under variation of the driving pulse peak intensity and detuning. The dipole phase shift of the autoionizing resonance is comparable to the case of the two-level system, but an asymmetry along the detuning axis arises from the weak-field asymmetric Fano absorption profile. More details are given in the text. In (**a**), the superimposed violet line indicates an unchanged dipole phase, $\Delta\varphi'_D \approx 0$ rad, whereas in (**b**), the jump around $q \to \pm\infty$, corresponding to a Lorentzian profile, is also marked in violet in the parameter region $I_0 = (3\ldots 4.7) \times 10^{15}$ W/cm$^2$ and $\Delta = (-0.65\ldots -0.2)$ eV.

Further, a jump from $q \to -\infty$ to $q \to +\infty$ is clearly visible at the borderline between black and white colors in Figure 4b in the region of higher intensities between $I_0 > 3 \times 10^{15}$ W/cm$^2$ and $I_0 = 4.7 \times 10^{15}$ W/cm$^2$ and red detunings between $\Delta = -0.65$ eV and $\Delta < -0.2$ eV, which is highlighted with another superimposed violet line. Thus, the weak-field Fano line shape can be turned into a Lorentzian shape with the correct choice of XUV pulse intensity and detuning. This agrees qualitatively with previous experimental findings for the modification of the helium 2s2p resonance at an XUV-FEL facility [16]. There we regard the strong coupling by the FEL pulse to be mainly driven by a few temporally short coherence spikes within the much longer average FEL pulse duration. The temporal duration of these coherence spikes relates to the average FEL spectral bandwidth and amounts to a few femtoseconds [17], which approximately corresponds to the Gaussian pulse duration (T = 3 fs) used in this work. A more quantitative investigation of the impact of the average FEL pulse duration on the observed line-shape asymmetry will be subject of a forthcoming work [26].

## 4. Conclusions

In summary, we have utilized a numerical two- and three-level system to investigate the influence of the near-resonant detuning on the resonant absorption line shape for a strongly coupled XUV transition to an excited or autoionizing state in the impulsive limit, respectively. We used the relation between the Fano q asymmetry parameter and the dipole phase shift [8] and linked the observed phase shift to energy level shifts of a strongly coupled resonant transition. We could therefore explain how the near-resonant detuning of a strong XUV driving pulse can change and control the observed absorption line shape. We verified this assumption with a comparison to an analytical approximation based on the generalized Rabi formalism which yields a very good agreement. With the help of this analytical approximation we concluded that the broadband spectrum of the XUV pulse cannot be neglected, which is inherently connected to its ultrashort temporal duration in the limit of impulsive dressing of resonant transitions. Spectrally averaging over the detuning-dependent shift of the analytical approximation agrees well with the numerical results. Our numerical model also reveals line-shape asymmetry changes for the autoionizing 2s2p resonance in helium driven directly from the 1s$^2$ ground state. Hereby, the line-shape modification and dipole phase shift induced by the intense XUV pulse reveals different trends for different detunings in an asymmetric manner. This asymmetry can be qualitatively explained by the weak-field limit of an initially asymmetric absorption profile. Notably, for a detuning around $-0.2$ eV the resonance line shape approximately keeps its weak-field asymmetry. We have shown in agreement with previous experimental results [16], that an intense and slightly red detuned XUV pulse may modify the absorption line shape of the weak-field Fano resonance to

a Lorentzian-like shape at high XUV intensity, with sensitivity to impulsively induced energy level shifts. Since our approach can be straightforwardly extended to any core-excited electron resonances in atomic targets, we expect these results to be of great interest for any spectroscopy and imaging experiments that are based on intense and short XUV and x-ray pulses generated from FEL facilities, where the average (X)FEL spectral bandwidth may become (much) broader than the natural spectral bandwidth of the coupled transition.

**Author Contributions:** Conceptualization, A.M. and T.P.; methodology, A.M., T.D. and L.A.; software, A.M. and L.A.; validation, A.M., L.A., T.D., P.R., M.R., C.O. and T.P.; formal analysis, A.M., L.A., T.D., P.R., M.R., C.O. and T.P.; investigation, C.O. and T.P.; data curation, A.M. and L.A.; writing—original draft preparation, A.M.; writing—review and editing, L.A., T.D., P.R., M.R., C.O. and T.P.; visualization, A.M. and C.O.; supervision, C.O. and T.P.; project administration, C.O. and T.P.; All authors have read and agreed to the published version of the manuscript.

**Funding:** This research received funding from the European Research Council (ERC) (X-MuSiC-616783).

**Conflicts of Interest:** The authors declare no conflict of interest.

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
