# Peer review of "Bound-State Electron Dynamics Driven by Near-Resonantly Detuned Intense and Ultrashort Pulsed XUV Fields"

_applsci, doi:10.3390/app10186153_

Round 1

Reviewer 1 Report

This paper follows up on a previous publication (ref. 1) and anticipates another (ref. 26). The paper describes a theoretical study of the absorption spectrum of He excited to a resonant, doubly excited state by an ultrashort (few fs), high intensity XUV pulse. The work has been carried out carefully and provides new insights into the nature of ultrafast, high field physics for this system, in particular by focusing on the dipole moment of the system. This work will be of interest to readers working in the fields of ultrafast and strong field physics, and I therefore recommend its publication in the present form, with the following optional minor corrections.

Line 139. “The states energies repel each other.. ” This is a rather colloquial description: in general energy does not repel energy. Is it more precise to say the states` energy separation increases with increasing XUV laser intensity?

Line 234. “monotonous” should be “monotonic”.

Line 239. There is no reason for FWHM to be in italics.

Lines 287, 288 are a bit confusing. Do you mean “resonance, but rather it is cancelled out for a slightly red detuned pulse, ..” ?

Line 308. “suspect” should be “subject”.

Line 311. The phrase “a numerical two- and three-level system” would read better as

“a numerical model of  a two- or three-level system”

Line 314. “the Fano-q-asymmetry-parameter”. Too-many-hyphens!

Line 332. “core-hole—excited-electron”. Too many hyphens again, and one is very big.

Reviewer 2 Report

The manuscript presents a short study investigating origin of the shape of photoabsorption lines arising in interaction of two- or three-level systems with strong and short XUV pulses. The authors use a direct numerical model comprising the two or three levels under consideration, accompanied by constants selected to mimic the helium 1s^2-2s2p(-continuum) excitation(-ionization) process, to asses the effect of the off-resonance detuning and peak intensity of the XUV pulse on parameters of the resonance line related to the double excited helium state 2s2p.

I consider the article well written and support its publication in the present form.

I have only two minor formal comments that may be addressed if there is an opportunity and wish:

  1. It is said that the dime-dependent Schrodinger equation is "... solved as an eigenvalue problem on a discrete time grid", which I find not sufficiently comprehensible. Maybe writing something like "... solved on a discrete time grid by finite-difference evolution in the eigenvector basis at each time step" would be more accurate.
  2. The Fano asymmetry parameter "q" is used in the text before it is introduced in Eq. (4); ideally, this should be interchanged.
